# Response Prediction in Immune Checkpoint Inhibitor Immunotherapy for Advanced Hepatocellular Carcinoma

**DOI:** 10.3390/cancers13071607

**Published:** 2021-03-31

**Authors:** Hao-Chien Hung, Jin-Chiao Lee, Yu-Chao Wang, Chih-Hsien Cheng, Tsung-Han Wu, Chen-Fang Lee, Ting-Jung Wu, Hong-Shiue Chou, Kun-Ming Chan, Wei-Chen Lee

**Affiliations:** 1Division of Liver and Transplantation Surgery, Department of General Surgery, Chang Gung Memorial Hospital at Linkou, Taoyuan 33357, Taiwan; mp0616@cgmh.org.tw (H.-C.H.); b9302012@cgmh.org.tw (J.-C.L.); b9002072@cgmh.org.tw (Y.-C.W.); chengcchj@cgmh.org.tw (C.-H.C.); domani@cgmh.org.tw (T.-H.W.); lee5310@cgmh.org.tw (C.-F.L.); wutj5056@cgmh.org.tw (T.-J.W.); Chouhs@cgmh.org.tw (H.-S.C.); chankunming@cgmh.org.tw (K.-M.C.); 2Medicine, Chang-Gung University College of Medicine, Taoyuan 33357, Taiwan

**Keywords:** NLR, hepatocellular carcinoma, immunotherapy, patient-generated subjective global assessment, progression-free survival

## Abstract

**Simple Summary:**

Hepatocellular carcinoma is the most common liver malignancy. In the population with an advanced stage of the disease, outcomes could be disappointed by treating with molecular targeting agents because of low treatment response rates. It has gained improving effects of immune checkpoint inhibitor as an emerging treatment for advanced HCC (Hepatocellular carcinoma). However, this novel treatment regimen is quite expensive; to select suitable patients prior to treatment is crucial in daily practice. Here, we intend to present the effect of immunotherapy in treating advanced hepatocellular carcinoma in the real world and to assess potential factors predicting treatment responses for patient selection.

**Abstract:**

Immune checkpoint inhibitors (ICI) have been applied to treat advanced stage hepatocellular carcinoma (HCC) and obtain promising effects. However, tumor response to treatment was unpredictable. A predicting biomarker of objective response or disease-control is an unmet need for patient selection. In this study, 45 advanced HCC patients who failed to sorafenib treatment and received nivolumab, 3 mg/kg bi-weekly, were included. Tumor responses to nivolumab treatment were assessed by the modified response evaluation criteria in solid tumors (mRECIST) criteria. Tumor responses were correlated to clinical characteristics to find out response predictors. In this small series, the prevalence of extrahepatic nodal metastasis, distant metastasis, and portal vein thrombus among the patients were 22.2% (*n* = 10), 48.9% (*n* = 22), and 42.2% (*n* = 19), respectively. The pre-treatment tumor size was 7.2 ± 4.2 cm in maximal diameter, and the calculated total tumor volume was 619.0 ± 831.1 cm^3^. Among 45 patients, 3 patients had partial response (PR), 11 had stable disease (SD), and the other 31 had progression of disease. By correlating clinical data to the patients with PR and SD, serum neutrophil-to-lymphocyte ratio (NLR) (hazard ratio (HR) = 2.04) and patient-generated subjective global assessment (PG-SGA) score (HR = 2.30) were the independent factors in multivariate analysis. By receiver operating characteristic curve analysis, pre-treatment NLR ≤ 2.5 and PG-SGA score < 4 were the cutoff points to predict tumor response to ICI treatment. In conclusion, biomarkers to predict tumor response for HCC are still lacking in this costly ICI therapy. In this study, NLR ≤ 2.5 and PG-SGA score < 4 indicated disease-control, and can be applied as biomarkers to select the right patients to receive this costly therapy.

## 1. Introduction

Hepatocellular carcinoma (HCC) is the major primary tumor in the liver [1]. Because tumors in the liver are always silent, many HCCs are already at advanced stage and inoperable once the tumors are found. According to the Barcelona Clinic Liver Cancer (BCLC) staging and therapeutic strategies, the patients at stage C should be treated systemically with molecular targeting agents or immune checkpoint inhibitors [2]. Molecular targeting agents have been applied to treat advanced HCC for many years and can prolong patients’ survival; however, tumor response rate to the treatment is only 1–3% [3,4,5]. Obviously, molecular targeting therapy is not enough for BCLC stage C patients. Currently, immune checkpoint inhibitor (ICI) is an emerging treatment for advanced HCC with promising effects [6].

The microenvironment of advanced HCC is infiltrated with high population of immunosuppressor cells such as regulatory T-cells and myeloid-derived suppressor cells which impair the function of CD8^+^ T-lymphocytes. Immune checkpoint inhibitors can block immunomodulatory pathways and rearm the function of impaired T-lymphocytes. Immune checkpoint inhibitors, anti-cytotoxic T-lymphocyte antigen-4 (CTLA4) or anti- programmed cell death protein 1 (PD-1)/anti-PDL-1, have been applied to treat advanced stage malignancy and obtain promising effects [6,7]. One-fifth of the metastatic melanoma patients could survive for more than 2 years when ipilimumab was applied [8]. When ipilimumab was combined with nivolumab for untreated melanoma, complete response rate could reach 22% and objective response rate was 61% [9]. Nivolumab and pembrolizumab were also applied to treat advanced stage non-small cell lung cancer and other malignancies with objective responses [10]. When nivolumab was applied to treat advanced HCC, 15% of response rate was achieved and nivolumab was approved to treat advanced HCC [6,11]. Although nivolumab is an anti-PD-1 monoclonal to block the PD-1/PDL-1 pathway, the expression of PD-1 on HCC is not correlated to treatment response. Until now, there are no available biomarkers to predict tumor responses for HCC under ICI immunotherapy. A predicting biomarker of objective response is unmet for patient selection when ICI is applied to treat advanced HCC patients.

Recently, combination therapy of ICI and anti-vascular endothelial growth factor (VEGF)/tyrosine kinase inhibitor (TKI) for advanced HCC improved the objective response rate to around 30% [12]. These results are exciting for the treatment of advanced HCC. However, who will respond to the treatment is still not predictable. In this study, we retrospectively reviewed the clinical profiles of the advanced HCC patients who received ICI to investigate whether we could find a clinical parameter to predict the clinical benefits for the patients and set a guidance for patient selection.

## 2. Results

### 2.1. Characteristics of Patients and Response to Nivolumab

A total of 45 patients who received nivolumab for advanced HCC were included in this study; 41 (91.1%) patients had liver cirrhosis but none of them was categorized as CTP (Child-Pugh classification) class C (the mean score was 5.3 ± 0.6). Most of the patients were male gender (*n* = 41, 91.1%), and the leading etiology was viral hepatitis (*n* = 38, 84.4%), followed by alcoholic hepatitis (*n* = 8, 17.8%). The period between HCC diagnosis and initiation of ICI treatment was 37.0 ± 32.5 months. Nineteen (42.2%) patients received curative hepatectomy and 17 (37.8%) patients had undergone loco-regional therapy, but the diseases recurred and progressed. Among all, three patients already had distant metastases at the time of HCC diagnosis. The average tumor size was 7.2 ± 4.2 cm in maximum diameter, and the calculated total tumor volume was 619.0 ± 831.1 cm^3^ prior to treatment. The prevalence of extrahepatic nodal involvement, distant metastasis, and PVT (Portal vein thrombosis) were 22.2% (*n* = 10), 48.9% (*n* = 22), and 42.2% (*n* = 19), respectively (Table 1). There were eleven (24.4%) patients who discontinued the immunotherapy treatment less than expected courses (4 experienced severe hepatitis, 1 died from drug-induced liver failure, 4 complicated with severe sepsis, 1 rapid deterioration of performance status beyond ECOG-PS (Eastern Cooperative Oncology Group-Performance Status) 2 and 1 expired due to massive EV (Esophageal varices) bleeding). However, all these patients had enough information to judge the response to immunotherapy. The median interval between the first dose and the radiological assessment was 66.0 (mean 74.3 ± 31.3) days. Among all 45 patients, 3 patients had PR (Partial response), 11 had SD (Stable disease) and the other 31 had PD (Progressive disease). Objective response rate was 6.67% and disease control rate was 31.1%. The best response for target lesions by each treated individual was presented, based on the maximal percentage of diameter change (Figure 1).

### 2.2. Difference between the Patients With or Without Disease-Control

The patients were divided into two groups according to the absence or presence of PD (non-PD group, *n* = 14 and PD group, *n* = 31). According to the treatment responses, a comparative study between non-PD and PD groups was analyzed and summarized in Table 2.

The mean serum NLR (Neutrophil-lymphocyte ratio) was significantly higher in the PD group compared to the non-PD group (4.4 ± 2.3 vs. 2.9 ± 1.3; *p* = 0.028). A higher proportion of baseline cirrhosis was observed in the PD group compared to the non-PD group (*n* = 30, 96.8% vs. *n* = 11, 78.6%; *p* = 0.058). The ALBI (Albumin-bilirubin score) and patient-generated subjective global assessment (PG-SGA) score were in high-levels in the PD group (both *p* < 0.005). There was no difference in age, gender, liver function tests, cirrhotic status, appearance of EV and ascites, albumin level, HCC etiology between the two groups. Regarding tumor-associated factors, tumor size, hilar nodal metastasis, distant metastasis, PVT, and AFP (Alpha-fetoprotein) were not different between the PD and non-PD groups, either.

### 2.3. Univariate and Multivariate Logistic Regression

To find out the important parameters, the available clinical factors were analyzed to predict PD. In univariate logistic regression, serum NLR, tumor size, cirrhosis, ALBI, and PG-SGA scores were considered as potential risks for PD with *p*-value < 0.100. The serum NLR (*p* = 0.025; hazard ratio [HR] = 2.04; 95% confidence interval [CI] = 1.10–3.80) and the PG-SGA score (*p* = 0.039; HR = 2.30; 95% CI = 1.04–5.09) remained independent risk factors for PD in multivariate analysis (Table 3).

### 2.4. Predictive Value of Serum NLR and PG-SGA

To unravel the predictive value of serum NLR and PG-SGA, ROCs (Receiver operating characteristic curve) of pre-treatment NLR and PG-SGA were performed. The AUROC (Area under receiver operating characteristic curve) of pre-treatment serum NLR in disease-control prediction were 0.774 (95% CI: 0.612–0.936, *p* = 0.004, Figure 2a). The optimal cut-off point was 2.52 with sensitivity/specificity at 57.1%/96.8%. The AUROC of pre-treatment PG-SGA in disease-control prediction was 0.747 (95% CI: 0.610–0.886, *p* = 0.009, Figure 2b). The optimal cut-off point was 3.5 with sensitivity/specificity at 100%/45.2%. In serial follow-ups during ICI immunotherapy, NLR was also checked at day14 (D14) after treatment initiation and post-treatment. The AUROCs of serum NLRs in disease-control prediction were 0.683 (95% CI: 0.514–0.853, *p* = 0.051 Figure 2c) at D14, and 0.770 (95% CI: 0.616–0.923, *p* = 0.001, Figure 2c) at post-treatment. The optimal cut-off values were 4.08 for 2-week after treatment initiation and 2.72 for post-treatment with sensitivity/specificity at 85.7%/58.1% and 64.3%/87.1%%, respectively.

### 2.5. Progression-Free Survival in Nivolumab-Treated Patients

The impact of high and low serum NLR on PFS was assessed according to optimal cut-off values before, during, and after immunotherapy. Patients with pre-treatment NLR > 2.5 showed a worse PFS than the patients with NLR ≤ 2.5 (*p* = 0.004, Figure 3a). Similar results were demonstrated in NLR at 2-week after treatment initiation and post-treatment. During treatment, the patients with pre-treatment NLR ≥ 4.1 showed a worse PFS than the patients with NLR < 4.1, *p* = 0.006, Figure 3b). After treatment, the patients with NLR > 2.7 showed a worse PFS than the patients with NLR ≤ 2.7 (*p*= 0.001, Figure 3c). For the patients with PR and SD, the PFS was 103.2 ± 24.9 days up to now, compared with 61.3 ± 24.6 days for the patients with PD (*p* < 0.001).

### 2.6. Immune-Related Adverse Effect (irAE) Profiles

Among 45 advanced HCC patients treated with nivolumab, 29 (64.4%) patients experienced at least one irAE (Table 4). Skin category (*n* = 13, 28.9%) was the most common irAE, followed by hepatobiliary irAE (*n* = 11, 24.4%). Median times from immunotherapy initiated to skin and hepatobiliary irAEs were 3.1 weeks (range 0.6–7.6) and 4.6 weeks (range 1.0–14.6), respectively. Besides that, four and five patients had grade 3 (>5, ≤10 times) and grade 4 (>10 times) aminotransferase increase, respectively. One patient eventually died for drug-induced liver failure. However, a background analysis revealed that there was no association between irAE and treatment responses.

## 3. Discussion

Advanced HCC can be treated by ICI and yielded a response rate between 15–20% [6]. However, the dilemma of ICI treatment for advanced HCC is unpredictability of treatment response under this costly treatment. There are no effective biomarkers to select right advanced HCC patients to receive ICI therapy until now. Several biomarkers such as the expression of PD-L1 and tumor mutational burden have been applied to predict tumor response to ICI treatment in melanoma, urothelial cancer, lung cancer, etc [13,14,15,16,17]. However, the expression of PDL-1 on HCC cancer cells was low and could not reflect the clinical response of the tumors [6]. Tumor mutational burden could not be applied as a biomarker of ICI therapy for HCC, either [18]. Therefore, it is always a puzzle for clinical physicians to advise advanced HCC patients to have ICI therapy. In this retrospective study, NLR and PG-SGA scores both were independent factors to predict tumor response to ICI therapy for HCC. Compared to the patients with PD, the patients with PR or SD had low levels of NLR and PG-SGA scores. NLR and PG-SGA scores both are easy-to-obtain clinical data, and can be easily applied as biomarkers to predict the tumor response of ICI treatment for advanced HCC patients.

Chronic inflammation is a risk factor of malignancy. NLR is already reported as a predictor of outcomes for malignancy including gastric cancer, lung cancer, breast cancer, colon cancer, etc. [19,20,21,22]. NLR is also a predictor of surgical outcomes for HCC and low NLR had a better overall survival [23,24,25]. Actually, anti-cancer immunity is mediated through T-lymphocytes. High levels of T-lymphocytes may have a high ability of anti-cancer. NLR was found important in ICI therapy in lung cancer. The patients with NLR ˂ 5 had a better tumor response than the patients with NLR ≥ 5 [26]. NLR could also be recognized as predictors of ICI therapy in other cancers [27,28,29,30]. In this study, the patients with NLR ≤ 2.5 prior to ICI treatment had a better chance of disease-control. After treatment, the patient with NLR ≤ 2.7 had better responses and better progression-free survival. Clearly, NLR is an important factor to assess whether the tumor will response to ICI therapy prior to treatment or after treatment. NLR at 2.5 is the best cutoff point to select patients to receive ICI treatment.

Not only NLR, but also PG-SGA score are the predictor of ICI therapy for advanced HCC patients in this study. Nutrition is important for the patients with advanced malignancy to receive chemotherapy. PG-SGA currently is the standard of patient assessment in oncology and other chronic catabolic condition [31]. In this study, the scores of PG-SGA were correlated to therapeutic responses of ICI. The patients with disease control had lower PG-SGA scores than the patients with progressive disease. High PG-SGA scores represent catabolism in ill patients and decreased immunity which decreases T-cell cytotoxic abilities. Although ICI blocked the negative signals in T-cells, the cytotoxic capacity of T-cells is not enough to kill cancer cells and achieve clinical benefits.

Selection of the right patients to receive ICI treatment for advanced HCC is a crucial issue, because ICI is costly. Several biomarkers have been reported for patient selection with melanoma or non-small cell lung cancer; however, all were not effective in HCC patients. For HCC patients, a 10% reduction of serum alpha-fetoprotein level at 4 weeks or lower serum NLR and PLR at 6 weeks after treatment were mentioned as a marker of patient selection with disease control [32,33]. However, these were post-treatment markers and could not be used to predict tumor response prior to treatment for patient selection. Zhang et al. mentioned that hemoglobin level, portal vein tumor thrombus, and Child-Pugh score were associated with hyperprogressive disease [34]. This might be applied to exclude the patients who were not suitable to have ICI treatment. Efforts were also put in developing molecular biomarkers for predicting immunotherapy response including specific gene mutations, metabolomic and transcriptional fingerprints regarding Wnt/β-catenin activation status [35]. However, Wnt/β-catenin pathway did not present in all HCC, and was limited its application in clinical practice. In this study, we found that advanced HCC patients with NLR ≤ 2.5 and PG-SGA score < 4 prior to ICI treatment had a better disease control rate in ICI therapy. NLR ≤ 2.5 and PG-SGA score < 4 could be used as biomarkers of patient selection for advanced HCC with ICI therapy.

This is a retrospective study to seek the possible biomarkers for selecting advanced HCC patient to achieve clinical benefits under ICI treatment. In this study, NLR and PG-SGA are both good indicators to predict disease-control for ICI treatment. Both NLR and PG-SGA are easily obtained data in our daily practice, therefore, can be apply to select advanced HCC patients to receive ICI treatment. However, further prospective studies are needed to verify NLR and PG-SGA as pre-treatment biomarker for ICI treatment. Furthermore, lymphocytes in peripheral blood are consisted of many types of T-cells, such as cytotoxic T-cells, helper T-cell, regulatory T-cells, etc. The simple frequency of lymphocytes cannot reflect the true cytotoxicity to tumor cells. Further analysis to identify specific T-cells may improve treatment efficacy.

## 4. Materials and Methods

### 4.1. Patients

The HCC patients who were treated by nivolumab at Linkou Chang Gung Memorial Hospital, Taiwan, from April 2019 to April 2020, were reviewed. The diagnosis of HCC was made through pathologic examination or laboratory and radiological assessments according to the American Association for the Study of Liver Diseases (AASLD) [36] or the European Association for the Study of the Liver (EASL) [37] guidelines. All included patients who failed in sorafenib treatment for metastatic HCC, had inoperable HCC with portal invasion, or were refractory to transcatheter arterial chemoembolization. None of the patients had meningeal metastasis, progressing brain metastatic lesions, long-term steroid use for autoimmune disease or received allogeneic organ transplantation. After excluding the patients who were lacking essential information for further analyses (*n* = 4; two expired before assessing immunotherapy response and another two missed pre-treatment laboratory data), 45 advanced HCC patients were finally enrolled into analysis. This study protocol was confirmed to the ethical guidelines of the 1975 Declaration of Helsinki and was approved by the institutional review board of Chang-Gung Memorial Hospital (IRB No. 201901682B0).

### 4.2. Nivolumab Administration

Nivolumab was administered at doses of 3 mg/kg bi-weekly, and a complete treatment contained 6 courses of nivolumab. The treatment would be terminated early if the patient could not tolerate nivolumab-associated side effects or was not willing to continue treatment. Immunotherapy-orientated clinical examinations started since the day that nivolumab was given. It contained regular drug-related toxicity evaluation and regular laboratory tests at an outpatient clinic from the first dose. The examinations were repeated prior to each treatment course. The response to treatment and tumor status were evaluated after nivolumab immunotherapy was accomplished or discontinued.

### 4.3. Assessment of Responses to Treatment

To determine whether ICI immunotherapy was beneficial for tumor control, tumor sizes were measured at the end of the treatment courses. Tumor sizes were assessed by either contrast-enhanced computed tomography (CT) or magnetic resonance imaging (MRI). The modified Response Evaluation Criteria in Solid Tumors (mRECIST) criteria for HCC were utilized to evaluate treatment response [1]. A typical arterially enhancing tumor at least 1 cm in the longest diameter with clear delineation was selected as the target lesion. On the contrary, an infiltrative HCC with ill-defined border might not be suitable to be considered as target lesion but non-target lesion for mRECIST. The disappearance of intra-tumoral arterial enhancement should be considered equivalent to treatment response. In addition, difficult-measurable lesions such as portal vein thrombosis (PVT) and porta hepatis lymph node were also selected as non-target lesions. In summary, for target lesions, complete response (CR) is defined as the disappearance of all lesions; partial response (PR) is considered as an over 30% reduction in the sum of lesion diameters; progressive disease (PD) is for at least a 20% increment in the sum of lesion diameters or any appearance of a new lesion; stable disease (SD) is between the PD and PR qualification. For non-target ones, CR equals to the disappearance of all lesions; PR/SD means the persistence of one or more lesions; PD represents for the appearance of at least one new lesion and/or progressive existing lesions. All images were reviewed and interpreted by professional radiologists.

### 4.4. Clinical Profiles

The patients’ general condition prior to nivolumab administration, standard hematology/biochemistry, HCC-associated disease entities, tumor markers, tumor staging, and detailed HCC diagnosis, and treatment process/duration/response were all recorded. All patients received complete evaluations including Child–Turcotte–Pugh score, albumin–bilirubin (ALBI) score, Eastern Clinical Oncology Group performance status (ECOG-PS), Patient-Generated Subjective Global Assessment (PG-SGA) [38], and staging according to the 8th edition of the American Joint Committee on Cancer (AJCC) tumor, node, metastasis (TNM) staging system [39]. Total tumor volume was calculated by a sum of top 3 intrahepatic tumor volumes on the pre-treatment radiologic image according to the ellipsoid formula [40]: tumor volume [cm^3^] = (length [cm]) × (width [cm])^2^ × 0.52. All cases with ascites were only in minimal amount by pre-treatment CT or MRI. The establishment of an existence of esophageal varices (EV) was confirmed by panendoscopy evaluation, whiles the appearance of cirrhosis was made by liver echography/CT.

### 4.5. Neutrophil-to-Lymphocyte Ratio

With regards to serum neutrophil-to-lymphocyte ratio (NLR) as a known predictive indicator for HCC prognosis, it was checked at three different time points including pre-treatment (on the day of first dose of nivolumab), 2-week after 1st course of treatment, and post-treatment.

### 4.6. Clinical Benefits of Treatment

The patients were considered to have clinical benefits when the tumors had CR, PR, and SD to treatment. The progression-free survival (PFS) was calculated from the initiation day of nivolumab treatment to radiologic evaluation of tumor progression.

### 4.7. Statistical Analysis

Quantitative and qualitative factors were expressed by mean values ± standard deviations and numbers with percentages, respectively. Pearson’s chi-square test was used to compare categorized variables between PD and non-PD group, while independent T-test for continuous parameters. Potential risk factors with *p*-value < 0.100 in univariate logistic regression analysis were further entered into multivariate model with backward selection to identify independent ones. The predictive ability was examined using the area under receiver operating characteristic curve (AUROC), and optimal cut-off points were set under the Youden index and presented along with given sensitivity, specificity, positive predictive value (PPV), and negative predictive value (NPV). The progression-free survival (PFS) comparison was evaluated by the Kaplan–Meier method. Two-tailed *p*-value less than 0.05 was considered to be statistically significant. Analyses were accomplished using SPSS Statistics version 24.0 (SPSS Incorporation, Chicago, IL, USA).

## 5. Conclusions

The costly therapy of ICI with or without combination of molecular targeting agents may induce tumor responses in certain advanced HCC patients, but biomarkers to predict tumor response are lacking. In this retrospective study, NLR ≤ 2.5 and PG-SGA score < 4 indicate disease-control, and can be applied as biomarkers to select the right patients to receive this costly therapy.

## Figures and Tables

**Figure 1 cancers-13-01607-f001:**
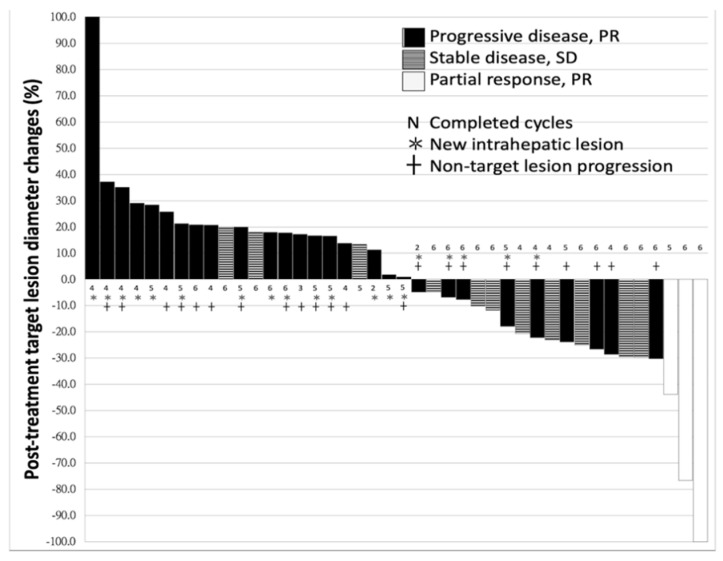
The waterfall plot of the change percentage in the sum of target lesion diameter (excluded 3 infiltrating HCC cases, without measurable target lesion) after cycles of treatment. The objective response rate was 6.67% and disease control rate was 31.1%.

**Figure 2 cancers-13-01607-f002:**
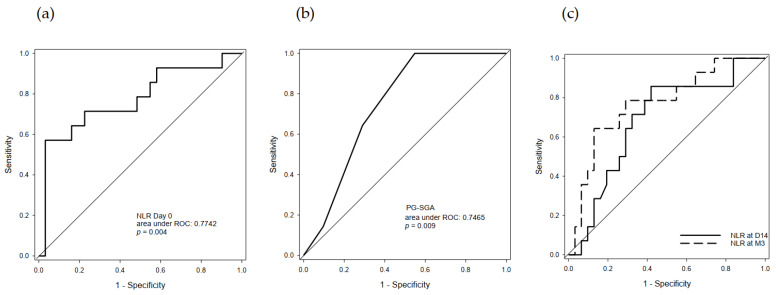
The receiver operating characteristic curve for NLR (Neutophil-lymphocyte ratio) and PG-SGA (Patient-generated Subjective global assessment). (**a**) The area under ROC (Receiver operating characteristic curve) for pre-treatment serum NLR in disease-control prediction were 0.774 (95% CI: 0.612–0.936, *p* = 0.004). The optimal cut-off point was 2.52 with sensitivity/specificity at 57.1%/96.8%. (**b**) The area under ROC of pre-treatment PG-SGA in disease-control prediction were 0.747 (95% CI: 0.610–0.886, *p* = 0.009). The optimal cut-off point was 3.5 with sensitivity/specificity at 100%/45.2%. (**c**) The area under ROC of serum NLR in disease-control prediction were 0.683 (95% CI: 0.514–0.853, *p* = 0.051) during treatment, and 0.770 (95% CI: 0.616–0.923, *p* = 0.001) at post-treatment. The optimal cut-off values were 4.08 for 2-week after treatment and 2.72 for post-treatment with sensitivity/specificity at 85.7%/58.1% and 64.3%/87.1%%, respectively.

**Figure 3 cancers-13-01607-f003:**
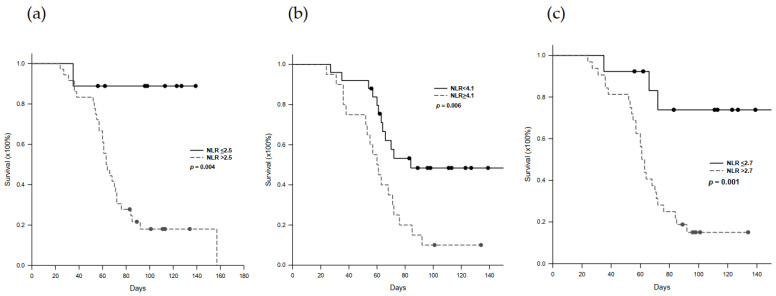
Progression-free Kaplan–Meier survival according to NLR. (**a**) Patients with pre-treatment NLR > 2.5 showed a worse PFS than the patients with NLR ≤ 2.5 (*p* = 0.004). (**b**) During treatment, the patients with pre-treatment NLR ≥ 4.1 showed a worse PFS than the patients with NLR < 4.1, *p* = 0.006). (**c**) After treatment, the patients with NLR > 2.7 showed a worse PFS than the patients with NLR ≤ 2.7 (*p* = 0.001).

**Table 1 cancers-13-01607-t001:** Baseline demographics of 45 advanced stage HCC patients underwent immunotherapy.

Factors	*n* = 45 (100%)
Baseline conditions	
Age (years)	61.8 ± 9.6
Gender (Male)	41 (91.1%)
CTP score	5.3 ± 0.6
ALBI score	−2.49 ± 0.39
Thrombocytopenia	11 (24.4%)
EV	7 (15.6%)
Ascites	10 (22.2%)
Cirrhosis	41 (91.1%)
ECOG-PS (0/1/2)	35/9/1
Viral hepatitis	38 (84.4%)
Alcohol consumption	8 (17.8%)
PG-SGA score	3.8 ± 2.9
Tumor-associated factors	
Maximum tumor diameter (cm)	7.2 ± 4.2
Total tumor volume (cm^3^)	619.0 ± 831.1
Alpha-fetoprotein (ng/mL)	82,584.2 ± 499,364.2
T stage (2/3/4)	13/12/20 (28.9/26.7/44.4%)
N stage (1)	10 (22.2%)
M stage (1)	22 (48.9%)
PVT	19 (42.2%)
Serum NLR	4.0 ± 2.2
Medical treatment	
Anti-viral agent	22 (48.9%)
Previous history of hepatectomy	19 (42.2%)
Previous Sorafenib	45 (100.0%)
Period between diagnosis and ICI (months)	37.0 ± 32.5
Early drop-out of ICI treatment	11 (24.4%)
Response to ICI (PR/SD/PD)	3/11/31 (6.7/24.4/68.9%)

HCC, hepatocellular carcinoma; CTP score, Child-Turcotte-Pugh score; ALBI, Albumin-Bilirubin; EV, esophageal varices; ECOG, Eastern Clinical Oncology Group performance status; PG-SGA, Patient Generated Subjective Global Assessment; PVT, portal vein thrombus; NLR, neutrophil-to-lymphocyte ratio; ICI, immune checkpoint inhibitor; PR, partial response; SD, stable disease; PD, progressive disease.

**Table 2 cancers-13-01607-t002:** A comparative study between the patients with or without disease control under ICI (Immune checkpoint inhibitor).

Factors	PR + SD (*n* = 14)	PD (*n* = 31)	*p*-Value
Age (years)	65.2 ± 10.2	60.3 ± 9.1	0.117
Gender (Male)	14 (100.0%)	27 (87.1%)	0.159
WBC (×10^9^/L)	6.1 ± 2.0	5.7 ± 2.2	0.571
Platelet(×10^3^/μL)	168.6 ± 76.3	193.4 ± 149.3	0.558
INR	1.1 ± 0.1	1.2 ± 0.1	0.163
NLR	2.9 ± 1.3	4.4 ± 2.3	0.028
PLR	123.7 ± 65.4	190.7 ± 118.6	0.185
Creatinine (mg/dL)	1.0 ± 0.2	0.9 ± 0.6	0.734
Total bilirubin (mg/dL)	0.7 ± 0.3	0.9 ± 0.5	0.184
AST (U/L)	62.3 ± 30.5	82.7 ± 54.7	0.200
ALT (U/L)	48.1 ± 29.2	63.5 ± 49.9	0.291
Albumin (g/dL)	4.0 ± 0.4	3.7 ± 0.4	0.059
CTP class (A/B)	14/0 (100.0/0.0%)	29/2 (93.5/6.5%)	0.331
ALBI score	−2.7 ± 0.3	−2.4 ± 0.4	0.042
EV	3 (21.4%)	4 (12.9%)	0.465
Ascites	2 (14.3%)	8 (25.8%)	0.389
Cirrhosis	11 (78.6%)	30 (96.8%)	0.047
ECOG-PS (0/1/2)	12/2/0 (85.7/14.3/0.0%)	23/7/1 (74.2/22.6/0.3%)	0.623
Viral hepatitis (Yes)	12 (85.7%)	26 (83%)	0.874
Alcohol use (Yes)	3 (21.4%)	5 (16.1%)	0.667
PG-SGA score	2.3 ± 0.7	4.7 ± 3.2	0.003
AFP (ng/mL)	3540.9 ± 5481.5	118281.2 ± 601217.5	0.482
Max. tumor diameter (cm)	5.6 ± 3.7	8.0 ± 4.3	0.091
Total tumor volume (cm^3^)	397.2 ± 659.3	719.1 ± 889.6	0.233
T (2/3/4)	6/5/3 (42.9/35.7/21.4%)	7/7/17 (22.6/22.6/54.8%)	0.110
N (1)	2 (11.1%)	8 (27.3%)	0.389
M (1)	7 (50.0%)	15 (48.4%)	0.920
PVT	4 (28.6%)	15 (48.4%)	0.213
Anti-viral agent	8 (57.1%)	14 (45.2%)	0.457
Period between diagnosis and IC (months)	35.4 ± 43.2	37.8 ± 27.1	0.822

HCC, hepatocellular carcinoma; ICI, immune checkpoint inhibitor; PR, partial response; SD, stable disease; PD, progressive disease; WBC, white blood cell; INR, international normalized ratio; NLR, neutrophil-to-lymphocyte ratio; PLR, platelet-to-lymphocyte ratio; AST, aspartate transaminase; ALT, alanine transaminase; CTP score, Child-Turcotte-Pugh score; ALBI, albumin-bilirubin; EV, esophageal varices; ECOG-PS, Eastern Clinical Oncology Group performance status; PG-SGA, Patient Generated Subjective Global Assessment; AFP, Alpha-fetoprotein; Max. maximum; PVT, portal vein thrombus.

**Table 3 cancers-13-01607-t003:** Univariate and multivariate logistic regression of significant factors to predict PD (Progressive disease) development after immunotherapy.

	Univariate			Multivariate		
	HR	95%CI	*p*-Value	HR	95%CI	*p*-Value
Serum NLR	2.07	1.10–3.90	0.025	2.04	1.10–3.80	0.025
Max. tumor size	1.17	0.97–1.40	0.099			
ALBI score	6.11	1.01–37.23	0.050			
Cirrhosis	8.18	0.77–87.20	0.082			
PG-SGA score	2.02	1.08–3.80	0.029	2.30	1.04–5.09	0.039

PD, progressive disease; HR, hazard ratio; CI, confidence interval; NLR, neutrophil-to-lymphocyte ratio; ALBI, Albumin-bilirubin; PG-SGA, Patient Generated Subjective Global Assessment.

**Table 4 cancers-13-01607-t004:** Immune-related adverse events according to category and grade.

Category	Total (*n* = 45)	Patients, N (%) ^a^ Grade 1–2	Grade 3–4 ^c^	Grade 5 ^b^	Weeks to Onset Median (Range)
Any	29 (64.4)	17 (37.7)	11 (24.4)	1 (2.2)	
Skin	13 (28.9)	13 (28.9)	0 (0.0)	0 (0.0)	3.1 (0.6–7.6)
Rash	6 (13.3)	6 (13.3)	0 (0.0)	0 (0.0)	
Pruritus	9 (20.0)	9 (20.0)	0 (0.0)	0 (0.0)	
Pneumonitis	4 (8.9)	1 (2.2)	3 (6.7)	0 (0.0)	8.3 (2.0–12.0)
Endocrine	1 (2.2)	1 (2.2)	0 (0.0)	0 (0.0)	6.0 (NA)
Thyroditis/hypothyroidism	1 (2.2)	1 (2.2)	0 (0.0)	0 (0.0)	
Hypophysitis	0 (0.0)	0 (0.0)	0 (0.0)	0 (0.0)	
Gastrointestinal	7 (15.6)	6 (13.3)	1 (2.2)	0 (0.0)	7.6 (0.6–8.9)
Mucositis	2 (4.4)	2(4.4)	0 (0.0)	0 (0.0)	
Esophagitis	1 (2.2)	1 (2.2)	0 (0.0)	0 (0.0)	
Diarrhea/colitis	5 (11.1)	4 (8.9)	1 (2.2)	0 (0.0)	
Hepatobiliary	11 (24.4)	2(4.4)	9 (20.0)	0 (0.0)	4.6 (1.0–14.6)
Hepatitis	9 (20.0)	2(4.4)	6 (13.3)	1 (2.2)	
Cholangitis	4 (8.9)	0 (0.0)	4 (8.9)	0 (0.0)	
Others	12 (26.7)	15 (33.3)	0 (0.0)	0 (0.0)	4.3 (1.0–10.3)
Fatigue	7 (15.6)	6 (13.3)	1 (2.2)	0 (0.0)	
Anorexia	7 (15.6)	7 (15.6)	0 (0.0)	0 (0.0)	
Polyarthritis	1(2.2)	1 (2.2)	0 (0.0)	0 (0.0)	

^a^ Percentages may add up to more than 100 because some patients experienced more than 1 adverse event. ^b^ The only treatment-related death was post-immunotherapy liver failure. ^c^ Eight patients required permanent cessation of nivolumab therapy because of intolerance of high grade immune-related adverse effects.

## Data Availability

The data presented in this study are available on request from the corresponding author.

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
