# Peer review of "Response Prediction in Immune Checkpoint Inhibitor Immunotherapy for Advanced Hepatocellular Carcinoma"

_cancers, 2021, doi:10.3390/cancers13071607_

Round 1

Reviewer 1 Report

In this manuscript, Hung H. and colleagues have introduced valid biomarkers to predict the tumor response to immune checkpoint inhibitors (ICI) in patients with advanced hepatocellular carcinomas (HCC). Nivolumab is a very promising ICI to treat advanced HCC; however, clinical data have demonstrated that tumor response to this inhibitor was widely heterogeneous. The variability of tumor responses to ICI prompts the urgency to identify biomarkers that precisely predict the immunotherapy response in HCC. Thus, enhancing and optimizing patient selection and reducing the cost. The authors have demonstrated that (1) serum neutrophil-to-lymphocyte ratio and (2) patient-generated subjective global assessment score is very promising predictors of ICI response in advanced HCC. The manuscript is well written and organized; however, here are some suggestions to improve the overall quality of this manuscript:

  • The overall quality of figures 1 and 2 is poor. The figures need better resolution. 
  • The author should add several sentences to offer more information about the anti-programmed death 1 monoclonal antibody.
  • The authors must consider citing and discussing the very recent published papers in this field, including
    (1)  X, Zhou Z, Zhang S. Predicting hyperprogressive disease in patients with advanced hepatocellular carcinoma treated with anti-programmed cell death 1 therapy. EClinicalMedicine. 2020 Dec 13;31:100673. doi: 10.1016/j.eclinm.2020.100673. PMID: 33554079; PMCID: PMC7846667.
    (2) Kwee SA, Tiirikainen M. Beta-catenin activation and immunotherapy resistance in hepatocellular carcinoma: mechanisms and biomarkers. Hepatoma Res. 2021;7:8. doi: 10.20517/2394-5079.2020.124. Epub 2021 Jan 7. PMID: 33553649; PMCID: PMC7861492.

Author Response

Dear Editor:

  We are pleased that we have the opportunity to revise our manuscript. We appreciate reviewers’ comments and suggestions. Point-to-point responses to reviewer 1 were described as following:

  1. The overall quality of figures 1 and 2 is poor.

Reply: The quality of figures are improved in the revised manuscript.

  1. The author should add several sentences to offer more information about the anti-programmed death 1 monoclonal antibody.

Reply: Two sentences are added in the beginning of 2nd paragraph in Introduction section.

  1. The authors must consider citing and discussing the very recent published papers in this field, including
    (1)  X, Zhou Z, Zhang S. Predicting hyperprogressive disease in patients with advanced hepatocellular carcinoma treated with anti-programmed cell death 1 therapy. EClinicalMedicine. 2020 Dec 13;31:100673. doi: 10.1016/j.eclinm.2020.100673. PMID: 33554079; PMCID: PMC7846667.
    (2) Kwee SA, Tiirikainen M. Beta-catenin activation and immunotherapy resistance in hepatocellular carcinoma: mechanisms and biomarkers. Hepatoma Res. 2021;7:8. doi: 10.20517/2394-5079.2020.124. Epub 2021 Jan 7. PMID: 33553649; PMCID: PMC7861492.

Reply: Both papers were cited and discussed in Discussion section.

Sincerely Yours,

Wei-Chen Lee

Reviewer 2 Report

In this study, the authors retrospectively reviewed the clinical profiles of the advanced HCC patients who received ICI to  investigate whether they could found a clinical parameter to predict the clinical benefits for the patients and set a guidance for patient selection. Overall, this work is interesting, and the manuscript is well written. However, there are some minor critiques which needs to be addressed

  1. What is the treatment related adverse effects? It would be better if the authors could make a table for the same
  2. Please include the individual characterization of response in a graphical representation by showing the mortality, last dose when patient off treatment, first complete response, last dose and first partial response
  3. What is the subsequent therapy taken by the patients after nivolumab progression?
  4. Please include the Race/Ethnicity information of the patients in the table 1. Also please add in discussion if there is any difference in NLR in terms of race/ ethnicity
  5. Whether the authors have calculated the platelet to lymphocyte ratio? Although this may be beyond the scope of this paper, it would be interesting if the authors add few points in the discussion.

Author Response

Dear Editor:

  We are pleased that we have the opportunity to revise our manuscript. We appreciate reviewers’ comments and suggestions. Point-to-point responses were described as following:

  1. What is the treatment related adverse effects? It would be better if the authors could make a table for the same

Reply: We added Table 4 and a new paragraph to detail immune-related adverse effects. 

  1. Please include the individual characterization of response in a graphical representation by showing the mortality, last dose when patient off treatment, first complete response, last dose and first partial response

Reply: we added a waterfall plot of the change percentage in the sum of target lesion diameter as new added Figure 1. The objective response rate was 6.67% (PR, n=3) and disease control rate was 31.1% (PR + SD, n=14). The only one patient who had full target lesion reduction was categorized into partial treatment response due to incomplete exptrahepatic metastasis remission.

  1. What is the subsequent therapy taken by the patients after nivolumab progression?

Reply: Ten cases received subsequent systemic therapies including chemotherapy with 5-fluorouracil, mitoxantrone, and cisplatin (n=4), atezolizumab plus bevacizumab (n=1), clinical trial with ADI-PEG 20 and modified FOLFOX6 (n=2), pembrolizumab (n=1), lenvatinib (n=1), and dendritic cell-based immunotherapy (n=1). Three patients received liver-directed treatment either transarterial embolization or external beam radiation therapy for intrahepatic lesions, and two received palliative radiation for distant metastases. The other 16 patients received individualized supportive care includes pain management, nutrition management, and symptom management.

  1. Please include the Race/Ethnicity information of the patients in the table 1. Also please add in discussion if there is any difference in NLR in terms of race/ ethnicity

Reply: The patients enrolled in this study are all Taiwanese.

  1. Whether the authors have calculated the platelet to lymphocyte ratio? Although this may be beyond the scope of this paper, it would be interesting if the authors add few points in the discussion.

Reply: There was no significant difference between the two groups in the pre-treatment platelet-to-lymphocyte ratio (mean: 123.7±65.4 in the PR+SD group v.s 190.7±118.6 in the PD group, p=0.185). We added this result in the table 2. Thrombocytosis and neutrophil recruitment may be act as a reflection of systemic inflammation and protumor microenvironment. However, the platelet count is also easily influenced by varying degrees of cirrhosis (high prevalence in the current study), and we consider PLR should be interpreted with caution especially in patients with portal hypertension.

Sincerely Yours,

Wei-Chen Lee

Round 2

Reviewer 1 Report

This current version (#2) of Hung et al is much improved.  The authors have largely addressed the comments/questions raised with discussion and the inclusion of new experiments/controls, which made this study stronger and better-suited for publication.